# Developmental changes in straight gait in childhood

**Saori Miyagishima**[1], **Hiroki Mani**[2]*, **Yui Sato**[1,3], **Takahiro Inoue**[4], **Tadayoshi Asaka**[5], **Naoki Kozuka**[6]

**1** Division of Rehabilitation, Sapporo Medical University Hospital, Hokkaido, Japan, **2** Faculty of Welfare and Health Science, Oita University, Oita, Japan, **3** Graduate School of Health Sciences, Sapporo Medical University, Sapporo, Hokkaido, Japan, **4** Department of System Pathology for Neurological Disorders, Brain Research Institute, Niigata University, Niigata, Japan, **5** Faculty of Health Sciences, Hokkaido University, Sapporo, Hokkaido, Japan, **6** Department of Physical Therapy, School of Health Sciences, Sapporo Medical University, Sapporo, Hokkaido, Japan

* mani-hiroki@oita-u.ac.jp

**Data Availability Statement:** All relevant data are within the paper and its Supporting Information files.

**Funding:** This work was supported in part by Japan Society for the Promotion of Science (JSPS)

## Abstract

### Background

Understanding typical gait development is critical in developing suitable physical therapy methods for gait disorders. This study investigated the developmental changes and controlling mechanisms of straight gait.

### Methods

We conducted an experimental procedure among 90 participants, including 76 typically developing children and 14 healthy adults. The children were divided according to age into 3–4, 5–6, 7–8, and 9-10-year age groups. We created two indices to quantify straight gait using the extrapolated center of mass (XCOM; goal index, $XCOM_G$ and actual progress index, $XCOM_P$), which were calculated and compared between the groups. Stepwise multiple regression was used to examine the effects of each gait variable on $XCOM_G$ and $XCOM_P$. To eliminate the effects of multicollinearity, correlation coefficients were calculated for all gait variables.

### Results

Both $XCOM_G$ and $XCOM_P$ decreased gradually with age and were significantly larger in the 3–4 and 5–6 year groups than in the adult group. Multiple regression analysis showed that step velocity, step width, and the coefficiente of variation (CV) of the step width had independent coefficients of variation for the $XCOM_G$, and the symmetry index of step time, step width, and the CV of the step width had independent CV for the $XCOM_P$. These variables were selected as significant variables. The results showed that meandering gait was more pronounced at younger ages. Furthermore, straight gait observed in adulthood was achieved by the age of 7.

KAKENHI Grant Numbers 18K17676, 19K19901. The funders had no role in study design, data collection and analysis, decision to publish, or preparation of the manuscript.

**Competing interests:** The authors have declared that no competing interests exist.

**Abbreviations:** XCOM, Extrapolated center of body's mass.

## Conclusion

Pace (step velocity) and stability (step width and CV of step width) may contribute to $XCOM_G$, which assesses the ability to proceed in the direction of the target. Stability and symmetry may contribute to $XCOM_P$, which assesses the ability to walk straight in one's own direction of progress. Physical therapists could apply these indices in children to assess their ability to walk straight.

## Introduction

Neurodevelopmental disorders such as developmental coordination disorder (DCD) impair gait and postural control. The parents of children with DCD and other developmental disorders often complain that their children cannot walk straight. A previous study focusing on the gait ability of children with DCD at ages 7–12 (young children) and ages 12–17 (older children) showed that the younger the child, the greater the variation in the body's center of mass (COM) velocity and antero-posterior and lateral acceleration [1]. Additionally, children with attention deficit hyperactivity disorder have been reported to have greater variability in speed, cadence, and stride length [2]. In children with neurodevelopmental disorders, the inability to walk straight and the tendency to fall are related to postural control in immature walking. However, no studies have examined the relationship between the ability to walk straight and postural control.

This is primarily because no studies have examined when and how meandering gait develops into a straight gait in childhood. To develop appropriate physical therapy for gait disorders, it is important to understand various aspects of typical gait development [3]. To the best of our knowledge, straight gait has not been clearly identified; therefore, it may be defined by various values regarding the trajectory of the center of mass (COM) and/or foot position [4]. Vertical and mediolateral fluctuations of the COM while walking are larger in those aged ≤4 years than in adults, and are larger in the anteroposterior direction at age ≤7 years [5]. Based on these results, the stability of COM control during walking was reported to improve up to the age of 7 years [5]. Recently, the extrapolated center of mass (XCOM) based on an inverted pendulum model has been used to assess stability during dynamic tasks such as walking [6]. This variable takes both the deviation and velocity of the COM into consideration. Thus, XCOM can indicate an index of stability, and may be linked to gait straightness. Stability has previously been characterized as the displacement of XCOM from the direction of travel during one gait cycle [7]. XCOM along the mediolateral axis was reported to have a strong negative correlation with ages 1–11 years, which is linked to changes in the step time-distance parameters of gait [7]. Although there is no clear definition of straight gait, to evaluate the straightness of the gait, the deviation from the straight line connecting the start to the target needs to be evaluated in addition to the displacement from the participant's actual direction of travel. Therefore, we devised two indices to quantify the ability to walk straight using XCOM: 1) deviation from the intended direction (G index) and 2) body sway with respect to the direction of progress (P index). The G index evaluates the ability to move straight toward the target; thus, smaller values are thought to reflect a better ability to capture the target and appropriately control the gait in the direction of the target. A smaller P index is thought to reflect appropriately control of the body sway in the direction of progress. Previous studies using the G index have not been reported. We created the G index with the expectation that it can be a new index for evaluating gait and postural control while an individual walks towards a goal.

Mani et al. [8] clarified the developmental process of gait in childhood using the five functional domains of the gait variables reported by Lord et al. [9]. They examined 189 elderly people with steady gaits to classify 16 gait variables into five functional domains using principal component analysis. These domains were pace (step length, step time, and swing time standard deviation [SD]), rhythm (step time, stance time, and swing time), asymmetry (left-right asymmetry of step time, stance time, and swing time), variability (step length SD, step velocity SD, step time SD, and stance time SD), and postural control (step width, step width SD, and mediolateral asymmetry of step length) [8]. Mani et al. reported that pace, rhythm, and asymmetry developed relatively quickly, reaching adult levels around the age of 7 years, while variability and postural control were still not mature by age 10 years. Examining the relationships between these five gait functional domains and the straight gait indices could help us understand what kind of control functions are required to walk straight.

Therefore, the purpose of the present study was to clarify the developmental changes in straight gait and the control mechanisms involved. We hypothesized that: (1) gait becomes more straight with age and reaches the adult level after the age of 7 years [5]; (2) functional domains of the gait stability, such as asymmetry and postural control, are associated with the ability to perform straight gait, and, the G index and P index are associated with different control functions.

## Methods

### Subjects

Ninety subjects, including 76 typically developing children (age 3–10 years) and 14 healthy adults participated in this study. The children were divided into a 3–4 (n = 25), 5–6 (n = 25), 7–8 (n = 14), and 9–10 year age groups (n = 12). Table 1 shows the physical characteristics of each group. The inclusion criteria were typical gross motor development (based on the developmental history of the child obtained from the guardian), the ability to understand an explanation of the experiment, and the ability to walk independently. The exclusion criteria were a history of orthopedic surgery or neurological disease. The study's objective, experimental procedure, and possible risks were fully explained to the subjects and their guardians in advance, both verbally and in writing, and written consent was obtained from the guardians. This study was approved by the ethical review boards of Sapporo Medical University and the Oita University Faculty of Welfare and Health Sciences (approval nos. 28-2-52, F200016).

### Experimental procedures and tasks

The subject was instructed to walk barefoot at a comfortable speed on a straight walking path toward a research assistant (who encouraged them forward) 6 m away. The walking path was created with a length of 4m and a width of 1.5m (Fig 1). The instructions were very simple:

**Table 1. Demographic data.**

|  | 3–4 years | 5–6 years | 7–8 years | 9–10 years | Adults |
|---|---|---|---|---|---|
|  | (n = 25) | (n = 25) | (n = 14) | (n = 12) | (n = 14) |
| Sex male | 14 | 13 | 8 | 7 | 6 |
| female | 11 | 12 | 6 | 5 | 8 |
| Age (years) | 4.1 ± 0.6 | 6.0 ± 0.6 | 7.8 ± 0.5 | 9.8 ± 0.5 | 22.8 ± 2.7 |
| Height (cm) | 101.6 ± 7.9 | 113.1 ± 5.8 | 124.4 ± 4.1 | 133.5 ± 6.9 | 167.1 ± 7.4 |
| Weight (kg) | 16.3 ± 2.7 | 20.3 ± 3.3 | 23.5 ± 1.4 | 27.7 ± 3.7 | 58.6 ± 7.6 |
| Leg length (cm) | 45.5 ± 3.9 | 53.7 ± 3.7 | 62.1 ± 3.7 | 66.3 ± 4.3 | 83.7 ± 4.7 |

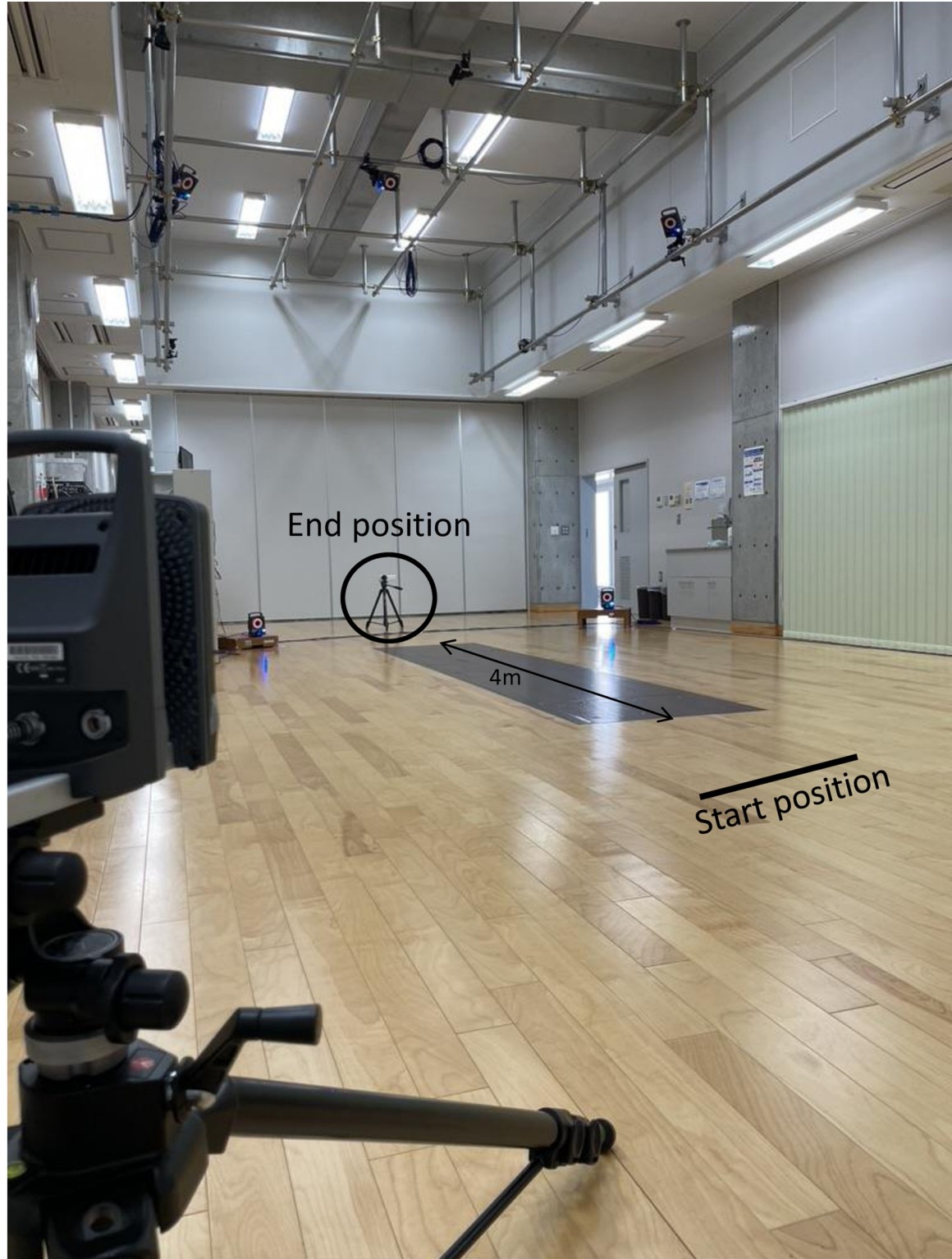

**Fig 1. Walking path.** The path was 4 m long and 1.5 m wide. It was surrounded by motion analysis system cameras. A research assistant stood at the end point along with a video camera.

"Please walk straight towards me." In some cases, parents gave instructions for younger children who could not follow the instructions of research assistants. Parents or research assistants stood behind video camera positioned at 1m away from end point, defined as the edge of walking path (Fig 1). Kinematic data were recorded during walking with a 10-camera VICON Nexus 3D motion analysis system (VICON, MX, USA). The sampling frequency was 100 Hz. A total of 27 infrared reflective markers 9.5 mm in diameter were fixed to the skin on the body and bone landmarks. The arrangement of the markers followed the method of Jensen [10] to calculate body-skeletal characteristics from physical measurement data. To eliminate the influence of acceleration and deceleration at the start and end of walking, we recorded the middle 4 m of the kinematic data. The start was always from the same position at 1m before from the edge of walking path (Fig 1). After practicing the task several times, the results were recorded five times. The mean value of these data was used for analysis. To eliminate the influence of fatigue, short breaks were taken at the subject's discretion, or after every two tasks. The length of the leg was defined as the distance from the greater trochanter to the floor.

## Data analysis

All signals were processed offline using MATLAB R2020b (MathWorks, Natick, MA, USA). The 3D data were processed using a Butterworth fourth-order low-pass filter (cutoff frequency, 20 Hz).

The analysis range for defining straight gait was 2 m, which was the maximum range over which all marker data for all participants could be recorded without missing data. At least two gait cycles were recorded per trial for all the subjects. Heel contact (HC) was defined as the time when the vertical coordinate of the calcaneal marker was at its minimum. The first HC was HC1st. COM was calculated using the body skeletal data from Jensen et al. [10]. In addition, XCOM was calculated using the equation of Hof et al. (Eq 1) [6], as follows:

$$XCOM = x + \frac{v}{\sqrt{g/l}} \tag{1}$$

where x is the COM position, v is the velocity of the COM, g is the gravitational acceleration, and l is the distance from the external malleolus marker to the COM.

Straight gait was quantified using the two methods (Fig 2). The G index was calculated using the XCOM coordinate at HC1st as the origin and was the integral of the deviation between the goal axis toward the end point, and the XCOM coordinate ($XCOM_G$). The larger this variable, the greater the deviation of the trajectory of the body from the axis from start to the goal. For the actual progressive index (P index), a straight line connecting the XCOM coordinates at HC1st and the XCOM coordinates at 2 m was defined as the progress axis. This was used to calculate the integral of the deviation between the progress axis and the XCOM coordinate ($XCOM_P$). The larger this variable was, the greater the postural sway with respect to the direction of travel, indicating less stability. These variables were normalized to the height of the COM [11].

Each gait variable was calculated using bilateral calcaneal markers and second metatarsal head markers. Foot off (FO) was defined as the time when the second metatarsal head marker exceeded 5% from the minimum to the maximum height. Step length (SL) was the anteroposterior distance between the heels from the HC on one side to the HC on the other side. The step width (SW) was defined as the distance between the heel markers on a perpendicular line connecting the directions of progress of each side. Step time (ST) was the time from the HC of one leg to the HC of the opposite leg. Step velocity (SV) was calculated by dividing SL by the step time. Stance time (STT) was defined as the time from the HC of one leg to the FO of the

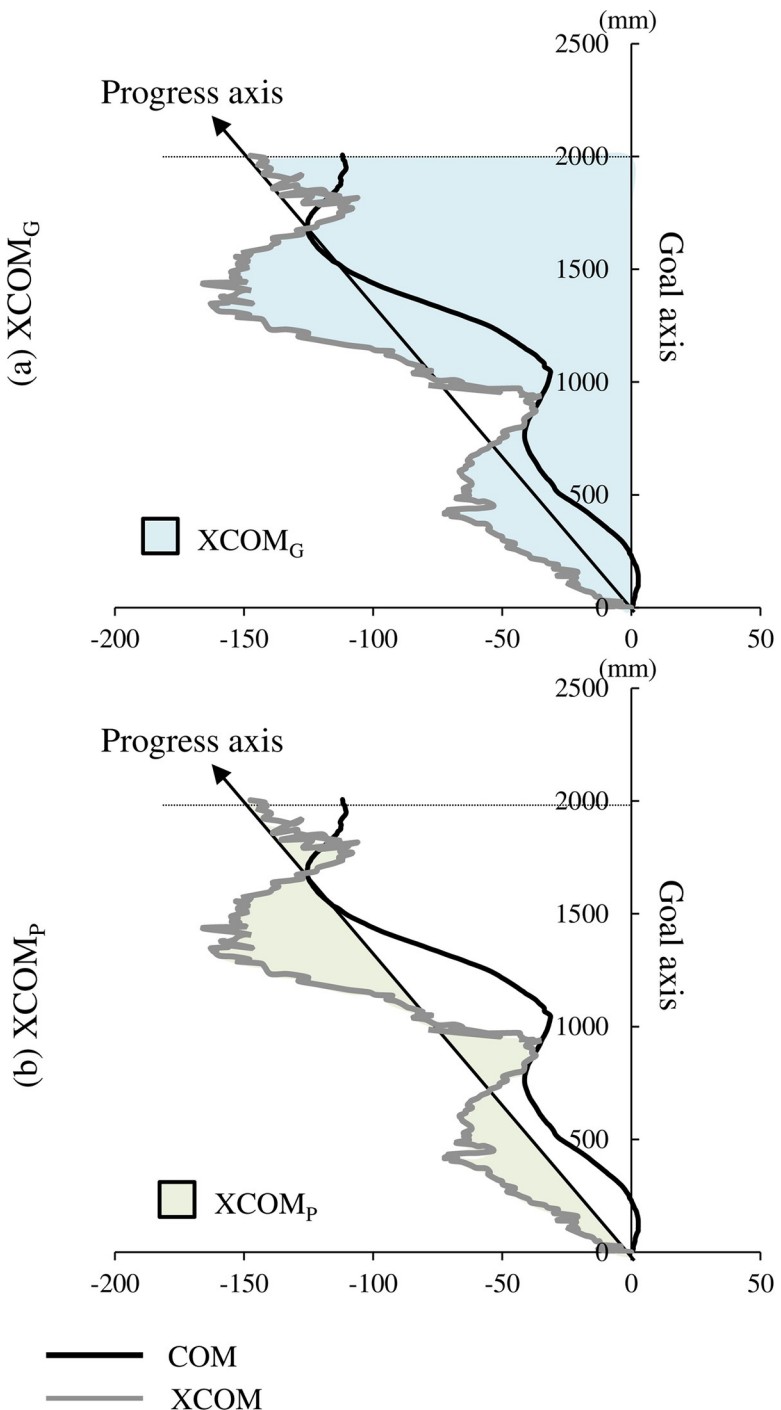

**Fig 2. Two indicators of straight gait; XCOMG and XCOMp.** Typical trajectories of the center of body's mass (COM) and the extrapolated COM (XCOM) in the horizontal plane. Black and gray lines represent the COM and XCOM, respectively. The goal axis is defined as the axis from start to end point, and the actual progress axis is defined as the straight line connecting the XCOM coordinates at HC1st and the XCOM coordinates at 2 m. $XCOM_G$ and $XCOM_P$ are calculated as the integral of the deviation between the XCOM coordinates and each axis. The blue area indicates the integral area of the $XCOM_G$, and the green area indicates the integral area of the $XCOM_P$.

same leg. Swing time (SWGT) was the time from the FO of one leg to the HC of the same leg. The means of the left and right steps were used as the representative values. For SD, the mean of the variance of the left and right steps was calculated (Eq 2) [12,13]. Lord et al. [13] used SD as a variable indicating variability, and the absolute values of the differences between each step were used as variables of left-right symmetry. However, because SD depends on the mean, the coefficient of variation (CV) is more suitable for analyzing variability. Moreover, the symmetry index (SI) has been shown to be a highly valid index of symmetry [14,15]. Therefore, in the present study, CV and SI were calculated as follows (Eq 3):

$$ SD = \sqrt{\frac{\text{variance of left step} + \text{variance of right step}}{2}} \tag{2} $$

$$ SI = \frac{|\text{left step} - \text{right step}|}{^{1}/_{2}}(\text{left step} + \text{right step}) * 100 \tag{3} $$

Distance variables such as SL and SW were normalized by dividing them by leg length (L), SV by ($\sqrt{gL}$), and time variables by the equation ($\sqrt{(L/g)}$) [16,17], where g indicates gravitational acceleration.

For the statistical analysis, all variables were normally distributed based on the Shapiro-Wilk test and the shape of the distributions with histograms. We performed a regression analysis to examine whether age affects $XCOM_G$ and $XCOM_P$ in typically developing children. ANOVA was used to compare the adult and child age groups, and the Tukey method was used for multiple comparisons. To examine the effects of velocity, as with SV, ANOVA was used to compare the adult and child age groups. Stepwise multiple regression was used to examine the effects of each gait variable on $XCOM_G$ and $XCOM_P$. To eliminate the effects of multicollinearity, correlation coefficients were calculated for all the gait variables. For variables with strong correlations ($r \geq |0.7|$), one was excluded from the analysis. Eight variables were used in the multiple regression analysis: pace: SL, SV; rhythm: ST; symmetry: ST_SI; variability: SL_CV, SV_CV; and stability: SW, SW_CV. The significance level was 0.05.

## Results

### Development of straight gait in childhood

Both $XCOM_G$ and $XCOM_P$ gradually decrease with age, and after the age of 7, the ability became equivalent to that of adults. Table 2 shows the results of the regression analyses of age with $XCOM_G$ and $XCOM_P$. Both $XCOM_G$ and $XCOM_P$ decreased significantly with development ($R^2 = 0.22$, $R^2 = 0.18$, $p < 0.01$) (Fig 3A and 3B).

Table 3 shows the mean values and SD for the $XCOM_G$, $XCOM_P$, and SV for the groups. $XCOM_G$ and $XCOM_P$ were both significantly different between the groups ($XCOM_G$: $F_{4,88} =$

**Table 2. Results of coefficient of regression.**

| a.$XCOM_G$ | Estimate | | Std. Error | Standard regression coefficient | p value |
|---|---|---|---|---|---|
| (Intercept) | 0.24 | | 1.99 | | < 0.01 |
| Age | -1.51 | | 0.32 | -0.473 | < 0.01 |

F-statistic: 21.85 on 1 and 76 DF, p-value < 0.01 $R^2 = 0.223$

| b.$XCOM_P$ | Estimate | Std. Error | Standard regression coefficient | p value | |
|---|---|---|---|---|---|
| (Intercept) | 0.157 | 1.036 | | <0.01 | |
| Age | -0.82 | 0.168 | -0.489 | <0.01 | |

F-statistic: 23.92 on 1 and 76 DF, p-value < 0.01 $R^2 = 0.178$.

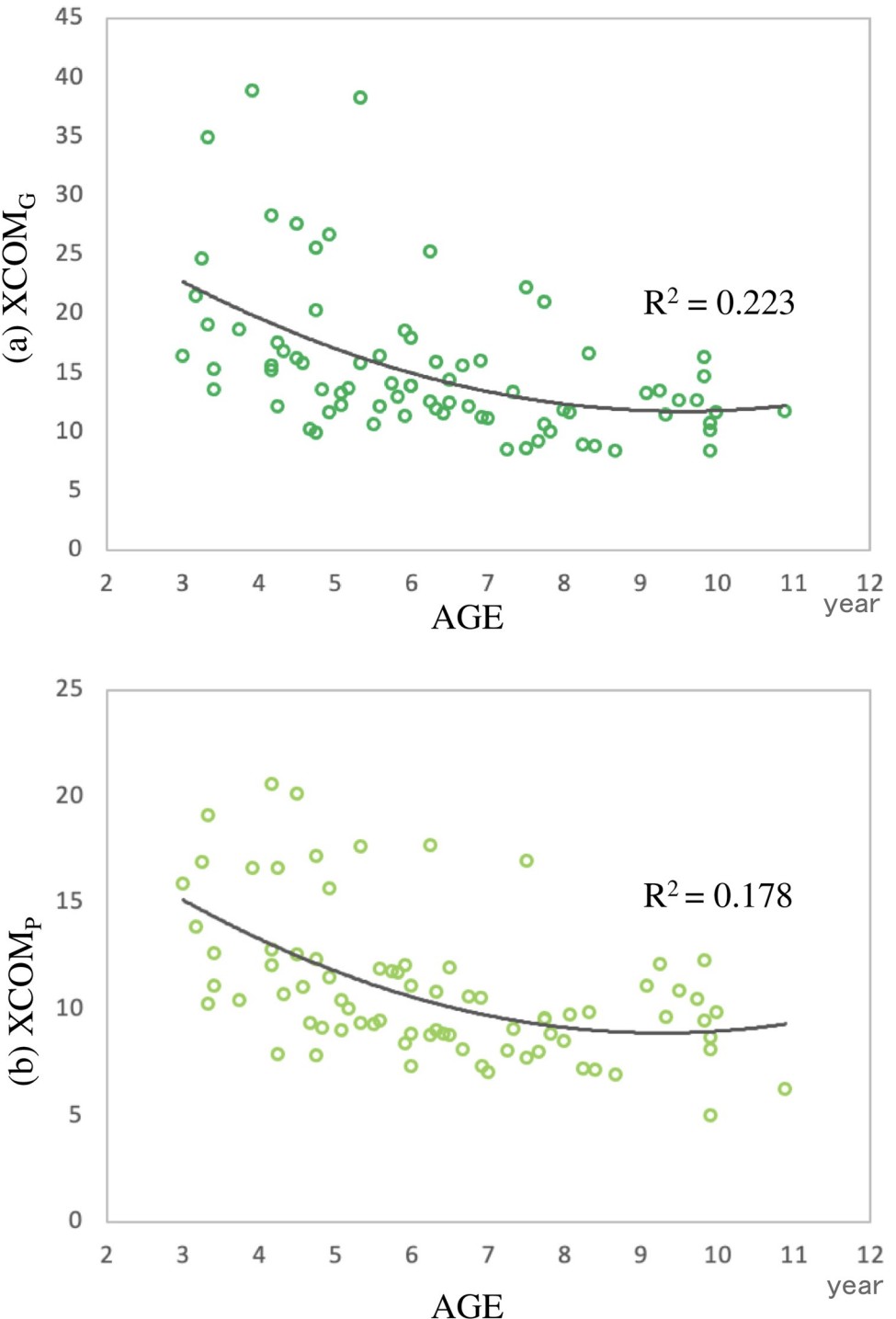

**Fig 3. Results of regression analysis performed for age.** (a) $XCOM_G$, and (b) $XCOM_P$.

11.85, $p < 0.01$; $XCOM_P$: $F_{4,88} = 14.14$, $p < 0.01$). The multiple comparison results showed that both $XCOM_G$ and $XCOM_P$ were significantly larger in the 3–4 and 5–6 year age groups than in adults. SV was not significantly different between the groups ($F_{4,88} = 1.57$, $p = 0.19$).

**Table 3. Results of XCOM and step velocity.**

|  | 3–4 years | 5–6 years | 7–8 years | 9–10 years | Adults |
|---|---|---|---|---|---|
| $XCOM_G$ | 19.5 ± 7.5* | 15.2 ± 5.6* | 12.1 ± 1.9 | 12.3 ± 2.1 | 8.3 ± 1.9 |
| $XCOM_P$ | 13.4 ± 1.7* | 10.4 ± 2.6* | 9.0 ± 2.5 | 9.5 ± 2.2 | 7.3 ± 0.9 |
| step velocity [%] | 49.3 ± 9.3 | 51.0 ± 7.1 | 48.6 ± 7.3 | 46.9 ± 5.6 | 45.8 ± 4.8 |

Mean ± SD. *: $p < 0.05$, compared to that of adults group (There were no differences between other groups).

## Control mechanisms associated with in straight gait

Table 4 shows the multiple regression results of $XCOM_G$ and $XCOM_P$ for each gait variable. For $XCOM_G$, SV (standardized regression coefficient [SRC] = 0.42, $p = 0.01$), SW (SRC = 0.61, $p < 0.01$), and SW_CV (SRC = 0.29, $p < 0.01$) were selected as independent significant variables ($R^2 = 0.69$). For $XCOM_P$, ST_SI (SRC = 0.19, $p = 0.02$), SW (SRC = 0.57, $p < 0.01$), and SW_CV (SRC = 0.38, $p < 0.01$) were selected as independent significant variables ($R^2 = 0.60$).

**Table 4. Result of multiple regression analysis.**

| a.$XCOM_G$ | Estimate | Std. Error | Standard regression coefficient | p value | |
|---|---|---|---|---|---|
| (Intercept) | 0.81 | 2.88 | | 0.78 | |
| Step length (SL) | 7.65 | 5.15 | 0.23 | 0.14 | |
| Step velocity (SV) | -0.18 | 7.37 | -0.42 | 0.01 | ** |
| Step time (ST) | 0.69 | 0.64 | 0.10 | 0.28 | |
| ST_SI | -0.13 | 2.64 | -0.003 | 0.96 | |
| SL_CV | -0.05 | 0.11 | 0.001 | 0.99 | |
| SV_CV | 8.93 | 5.04 | 0.16 | 0.08 | |
| Step width (SW) | 4.54 | 6.02 | 0.61 | <0.01 | ** |
| SW_CV | 8.31 | 2.32 | 0.29 | <0.01 | ** |

F-statistic: 23.13 on 8 and 84 DF, p-value < 0.01 $R^2 = 0.69$

| b.$XCOM_P$ | Estimate | Std. Error | Standard regression coefficient | p value | |
|---|---|---|---|---|---|
| (Intercept) | 1.09 | 6.22 | | 0.86 | |
| Step length (SL) | 11.19 | 11.16 | 0.17 | 0.32 | |
| Step velocity (SV) | -29.1 | 15.96 | -0.34 | 0.07 | |
| Step time (ST) | -0.19 | 1.39 | -0.02 | 0.89 | |
| ST_SI | 13.27 | 5.72 | 0.19 | 0.02 | ** |
| SL_CV | -20.67 | 23.38 | -0.06 | 0.38 | |
| SV_CV | 10.08 | 10.91 | 0.09 | 0.35 | |
| Step width (SW) | 81.26 | 13.03 | 0.57 | <0.01 | ** |
| SW_CV | 21.02 | 5.04 | 0.38 | <0.01 | ** |

F-statistic: 15.73 on 8 and 84 DF, p-value < 0.01 $R^2 = 0.60$

SI: Symmetry index, CV: Coefficient of variation.

## Discussion

### Development of straight gait in childhood

Overall, our results supported our first hypothesis; both the $XCOM_G$ and $XCOM_P$ scores decreased significantly with increasing age in children aged 3–10 years. Furthermore, comparison of the indices between adult and various age groups revealed that the adult level was reached at the age of 7 years. This result supports that of Dierick et al. [5], who found that stability improves up to the age of 7 years. The central nervous system controls posture during walking via complex processing that integrates input from the vestibular, visual, and somatic sensations [18]. Sensory information processing improves with development [19]. In childhood, when connectivity at the cortical-spinal level is immature, control via the spinal cord-brainstem is thought to contribute to the control of gross movements [20,21]. The central pattern generators (CPGs) associated with locomotion contribute to adjusting the motion patterns and rhythms of the limbs during walking [21]. Afferent information from the skin strongly affects CPGs [20], and somatic sensory function has been shown to reach an adult level at the age of 3–4 years [19]. This suggests that domains that express gait patterns, such as step, time factors, and left-right symmetry, are associated with corrections at the spinal cord-brain stem level, or CPGs, and are acquired relatively early (by about 7 years of age). Compared to the that of the adult group, the $XCOM_G$ and $XCOM_P$ results did not show statistically significant differences between the 7–8 years old and 9–10 years old groups in this study. Therefore, the development of straight gait continues until the age of 7 years, when a certain degree of gait control and various forms of posture control have been acquired. Although not significantly different, the two groups tended to have higher values than the adults, which may indicate that they develop gradually even after the age of seven.

### Control mechanisms involved in straight gait

Here, two indices were devised to capture meandering gait: 1) deviation from the direction toward the target (G index); and 2) body sway with respect to the direction of progress (P index). The G and P indices were found to be associated with different functional gait domains, supporting our second hypothesis.

XCOM_G evaluates the ability to go straight toward the target, and in addition to stability, it reflects the ability to accurately focus on the target and control the gait appropriately in that direction. Thus, this index expresses one component of the ability to walk straight. The $XCOM_G$ was significantly higher in the the 3–4 years old and 5–6 years old groups compared to that of the adult group. Therefore, the present results suggest that the ability to accurately focus on the target and control the gait appropriately in a particular direction reaches an adult-like level at age 7–8.

The results of this study indicate that SV, SW, and SW_CV contribute to $XCOM_G$ (Table 4). SV is classified under pace in the five functional gait domains. Because the present study used variables normalized by the height of COM, it eliminated the effects of physique, and thus likely reflects the growth of the parts of the nervous system involved in the basic control of walking [17]. Moreover, a correlation was observed between the SW and SW_CV. Both of these are classified as 'stability' in the five functional gait domains [8,9]. This indicates that the goal index is not only associated with the control mechanism of stability, but also with the basic mechanism for adjusting gait (pace). However, in a previous study, the authors found that SW in children aged 3–4 years was not significantly different from that in adults, though SW_CV in children up to the age of 9 or 10 years was significantly different from that in adults [9]. This suggests that the variable contributing to stability requires a high level of control that

is not mature even by the age of 10 years [9]. Gait stability requires controlling purposeful motions through predictive adjustment and constantly integrating various forms of sensory information generated during motion [18]. It has been suggested that this continues to develop even after the age of 10 years [9]. These results suggest that $XCOM_G$ is associated with SV, which is classified under pace, and with SW and SW_CV, which are classified under stability, and decreases gradually over time to accompany the growth of basic mechanisms for adjusting gait and the development of gait stability.

The multiple regression analysis conducted in this study indicated that ST_SI, SW, and SW_CV are involved in $XCOM_P$. Interestingly, unlike the G index, the P index was not only associated with stability functions, but also with ST_SI or symmetry functions. Left-right differences in spatiotemporal gait variables affect the deviation of COM during walking [22]. However, symmetry develops relatively early, reportedly reaching adult levels around the age of 3–4 years [9]. In contrast, SI variability is greater at younger ages [9]. $XCOM_P$ is designed to evaluate whether a person can maintain a straight line in the direction of intended progress, that is, the ability to appropriately control body sway. Therefore, the development of stability control to limit body sway and left-right symmetry control as it relates to stability are important to $XCOM_P$.

## Limitations

This study has several limitations which should be mentioned. First, we did not take sex differences into account, although prior research has found that sex influences postural control during walking [23]. Jensen et al.'s data [10], which we used in the analysis of COM, are body measurements from boys aged 4–15 years. The subjects of the present study included children as young as 3 years old and girls of various ages, which may have influenced the results. Secondly, we could not sufficiently examine data validity. There are no previous studies that investigated meandering gait; thus, we could not examine criterion-related validity. Moreover, the use of the XCOM measures for a whole trial (multiple gait cycles) as opposed to within step were different slightly to those stated by Hof et al. [6] and Hallemans et al. [7]. However, XCOM is a widely used analytical method, and we believe that because of its strong association with stability, it will not have a major impact on the conclusions of this study of novel indices. As such, more valuable results may be obtained by increasing the number of subjects and examining sex differences and validity.

## Conclusion

In this study, we devised two indices for straight gait and analyzed the developmental process in boys and girls aged 3–10 years. Stability and pace may contribute to the goal index, which assesses the ability to proceed in the direction of the target. Stability and symmetry may contribute to the ability to walk straight in one's own direction of progress, even if it is skewed. The two indices presented in this study are effective parameters for evaluating meandering and postural stability during gait.

## Supporting information

**S1 Data.**
(XLSX)

## Author Contributions

**Conceptualization:** Saori Miyagishima, Hiroki Mani, Yui Sato.

**Data curation:** Saori Miyagishima, Hiroki Mani, Yui Sato, Takahiro Inoue.

**Formal analysis:** Saori Miyagishima, Hiroki Mani.

**Funding acquisition:** Saori Miyagishima, Hiroki Mani.

**Investigation:** Saori Miyagishima, Hiroki Mani, Yui Sato, Takahiro Inoue.

**Methodology:** Saori Miyagishima, Hiroki Mani, Tadayoshi Asaka, Naoki Kozuka.

**Project administration:** Saori Miyagishima, Hiroki Mani.

**Resources:** Saori Miyagishima, Hiroki Mani.

**Software:** Saori Miyagishima, Hiroki Mani.

**Supervision:** Saori Miyagishima, Hiroki Mani, Yui Sato.

**Validation:** Saori Miyagishima, Hiroki Mani.

**Visualization:** Saori Miyagishima, Hiroki Mani.

**Writing – original draft:** Saori Miyagishima.

**Writing – review & editing:** Saori Miyagishima, Hiroki Mani, Tadayoshi Asaka, Naoki Kozuka.

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
