## [Decision Letter · Decision Letter 0]

7 Sep 2022

PONE-D-22-20977Developmental changes in straight gait in childhood aged 3-10 yearsPLOS ONE

Dear Dr. Mani,

Thank you for submitting your manuscript to PLOS ONE. After careful consideration, we feel that it has merit but does not fully meet PLOS ONE’s publication criteria as it currently stands. Therefore, we invite you to submit a revised version of the manuscript that addresses the points raised during the review process.

We look forward to receiving your revised manuscript.

Kind regards,

Laura-Anne Marie Furlong

Academic Editor

PLOS ONE

Journal Requirements:

"This work was supported in part by a Japanese Grant-in-Aid for Scientific Research (18K17676, 19K19901)."

Reviewers' comments:

Reviewer's Responses to Questions

**Comments to the Author**

1. Is the manuscript technically sound, and do the data support the conclusions?

Reviewer #1: Yes

Reviewer #2: Yes

2. Has the statistical analysis been performed appropriately and rigorously? 

Reviewer #1: I Don't Know

Reviewer #2: Yes

3. Have the authors made all data underlying the findings in their manuscript fully available?

Reviewer #1: Yes

Reviewer #2: No

4. Is the manuscript presented in an intelligible fashion and written in standard English?

Reviewer #1: Yes

Reviewer #2: Yes

5. Review Comments to the Author

Reviewer #1: The aim of this paper was to investigate the age related changes in, or development of, straight gait.

The authors used goal index and actual progress index to examine and quantify this.

The authors state they developed both indices but it is not clear if the P-index is as previously used in studies?

I have concerns regarding the G index as a measure of stability. The p-index makes sense and is a measure of body sway from the direction of progress.

The G-index than also incorporates deviation between the direction of progress and the target direction. The authors conclude that pace and stability may contribute to the g-index. However, as mentioned by the authors this measure also "reflects the ability to accurately focus on the target...". This potentially incorporates a large array of variables that have not been considered, some of which may also change with age (attention, ability to follow commands, eye-sight, other distractions in the room etc..). Its not clear from the Methods what instructions were given, how definite these instructions were and what the target was. All of this may impact of the g-index. it is also not clear what this index adds in comparison to just looking at the deviation of one, appropriate VICON marker from the A-P axis for example?

The findings in terms of the p-index confirm previous findings that stability is reached at about age 7.

Other points:

Abstract:

Line 30: typically developing might be better than normally developing

Line 39: CV not defined on first use

Introduction

Line 59: Need a reference for statement in first sentence

Line 69: Spell error- 'clearly'

Line 77 and line 69: I think 'meandering gait' and 'straight gait' need to be more clearly defined and then use one term consistently (if they are same thing?)

Methods:

more information needed on instructions and gait procedures given the young population and that walking towards target is one of the measured variables. How much instruction? What target? etc..

The middle 4 meters was recorded but then 2m was used for defining straight. Why only 2/4 and which 2?

Line 159: HC is presumably heel-contact but needs to be defined on first use

Line 195: don't think SD is defined on first use

Table 3: DEfine SV in legend (or table). p<0.05 is shown for differences compared to adult group but not clear if there were differences between other groups also or not?

Table 4: difficult to follow with all abbreviations and full variable names could easily be included

Discussion

Line 282: Am not sure it can be said that SV,SW etc contribute to the XCOM variables? They correlate with them which is different. This is maybe a bigger point that needs discussion and is a concern. The two XCOM variables are correlated with a host of other variables all (or at least many) of which are possibly age related so a correlation is not surprising and not sure what it tells us? If XCOM-G and XCOM P were correlated with height for example there may well be a correlation but that just reflects that both change with age/growth?

Reviewer #2: Developmental changes in straight gait in childhood aged 3-10 years

This is an interesting piece of research which explores meandering gait in childhood and adds more data to the growing field of children's gait biomechanics. I would however recommend some re-positioning of the article in terms of the introduction and rationale to make the justification of the research clearer.

Positioning and rationale of the study

The specificity of this statement to Japan sold be introduced in the first sentence. This also requires further referencing to justify the latter statement, as does the prevalence of low birth weight and it’s relationship to meandering gait.

The link between DCD and meandering gait is confusing – is this research driven and justified by children with DCD or those with meandering gaits? The inclusion criteria suggest that those with typical gait development were included only and therefore this meandering gait is typical in younger ages and not driven by birth weight or DCD, but just related to age and typical development.

The second paragraph of the introduction appears to be method heavy as opposed to introducing the topic. In it’s current format I am not sure that the introduction fits the rest of the manuscript, more references and description of typically developing gait and the development of stability would feel like it introduces the topic and additionally adds rationale to the participants and age groups being compared.

The purposes and hypothesise are confusing, particularly purpose 2.

Methods

The methods are generally clear and provide enough detail to be repeatable.

The use of the XCOM measures for a whole trial (multiple gait cycles) as opposed to within step appears to be different to the intended use by Hof (2005) and Hallemans et al. (2018), this is not clear within the introduction and has not been fully justified within the manuscript.

The descriptions of the variables XCOMg and XCOMp describe an integral, however this is not apparent in Figure 1. It looks like you have tried to take a single point in time as an example, but the figure could be improved and/or described properly. Furthermore the ‘Goal axis’ should be added to this figure along x = 0.

The latter descriptions of these indies on line 277-279 are far clearer than any earlier description so I would suggest using this terminology for the method and the figure to support the understanding of the terms.

The variables were defined over a period of 2 m with at least 2 gait cycles recorded:

How were these selected from the 6 m walkway?

Due to the evident range in COM position and XCOM over the gait cycle (from figure 1), should the number of gait cycles analysed for each individual have been standardised?

Results, discussion and conclusions

The Results section is clear with well-presented tables, but would benefit from a statement of results prior to the table introduction.

Again the discussion section would benefit from an introductory sentence or two which summarise the study findings prior to a sub-heading.

Line 257: is this supported by both of these variables or XCOMg only?

I don’t think the second and third hypothesis are addressed directly in the discussion.

The statement which concludes your paper: “The two indices presented in this study are effective parameters for evaluating meandering and postural stability during gait”. This feels like it requires further justification and presentation of data. To link with the introduction, gait which is defined as meandering should have been assessed and defined using your two parameters to compare to gait which isn’t meandering.

Minor comments:

Line 115: suggest ‘reduces to’

Line 117: ‘involved’ doesn’t make sense her as a term

Line 119: what are control mechanisms?

Line 132: who provide written consent? The guardian or the child?

Line 158: two gait cycles per trial or in total?

Line 220: I don’t think a % is an appropriate level and this should be reported as a decimal.

Line 258: please clearly define which age groups this statement is true for.

Table 1: sex should read ‘male’ or ‘female’

Table 3: check title –

Figure 1 title should define the two factors fully in the title.

6. PLOS authors have the option to publish the peer review history of their article (what does this mean?). If published, this will include your full peer review and any attached files.

Reviewer #1: No

Reviewer #2: No

---

## [Author Response · Author response to Decision Letter 0]

17 Oct 2022

We thank the reviewers for the meticulous review of our manuscript and the thoughtful comments. We have tried our best to address all the points raised. We have provided point-by-point responses to the reviewers’ comments. In the manuscript, the modified parts are shown in red.

Response to Reviewers:

Reviewer #1: 

The authors state they developed both indices but it is not clear if the P-index is as previously used in studies?

I have concerns regarding the G index as a measure of stability. The p-index makes sense and is a measure of body sway from the direction of progress.

The G-index than also incorporates deviation between the direction of progress and the target direction. The authors conclude that pace and stability may contribute to the g-index. However, as mentioned by the authors this measure also "reflects the ability to accurately focus on the target...". This potentially incorporates a large array of variables that have not been considered, some of which may also change with age (attention, ability to follow commands, eye-sight, other distractions in the room etc..). Its not clear from the Methods what instructions were given, how definite these instructions were and what the target was. All of this may impact of the g-index. it is also not clear what this index adds in comparison to just looking at the deviation of one, appropriate VICON marker from the A-P axis for example?

Response: We deeply appreciate the valuable comments of the reviewer. As far as we know, there is no previous research focused on the P index, but we believe that this should be a completely novel index. The G index is as described in the manuscript and is based on previous research. As you suggest, the G-index also incorporates the deviation between the direction of travel and the direction of the target, and is influenced by various factors. We have modified the purpose and hypothesis of this research more clearly (Line 102), and added detailed instructions to the methods section (Line 127). 

It is clear that various factors are involved in postural control. Therefore, it is required to evaluate straight progress including that element. We believe that it is not possible to consider various factors of attitude control by simply looking at the deviation of one suitable VICON marker from the AP axis.

other points: 

Abstract:

Line 30: typically developing might be better than normally developing

Response: Thank you for your advice. As you pointed out, I changed it to “typically”. (Line 30)

Line 39: CV not defined on first use

Response: Thank you for pointing this out. As you said, I defined the first "CV". (Line 39)

Introduction

Line 59: Need a reference for statement in first sentence

Response: Thank you very much for your pointing out. I decided that this part was not appropriate and deleted it.

Line 69: Spell error- 'clearly' 

Response: Thank you very much for your pointing out. This has been corrected accordingly. (Line 63)

Line 77 and line 69: I think 'meandering gait' and 'straight gait' need to be more clearly defined and then use one term consistently (if they are same thing?) 

Response: Thank you for your appropriate remarks. Since "meandering" and "straight" are confusing, I decided to use the term "straight gait" as a unified term.

Methods:

more information needed on instructions and gait procedures given the young population and that walking towards target is one of the measured variables. How much instruction? What target? etc..

Response: The explanation was certainly inadequate. As you pointed out, I added detailed instructions to methods (line 127).

The middle 4 meters was recorded but then 2m was used for defining straight. Why only 2/4 and which 2?

Response: Due to the experimental environment, we had no choice but to limit the analysis range for defining straight movement to 2m. The reason is that the maximum range for recording all marker data of all participants was 2m. I wrote about it in Line 148.

Line 159: HC is presumably heel-contact but needs to be defined on first use

Response: Thank you for your advice. ``Heel contact'' was added. (Line 150).

Line 195: don't think SD is defined on first use

Response: This has been defined in introduction section (Line 93).

Table 3: DEfine SV in legend (or table). p<0.05 is shown for differences compared to adult group but not clear if there were differences between other groups also or not?

Response: Thanking for pointing that out. It is indicated as “Step velocity” in the table. In addition, I added below the table that there was no difference from the non-adult group.

Table 4: difficult to follow with all abbreviations and full variable names could easily be included

Response: Thanks to your point, I defined the abbreviations other than SI and CV in the table, and defined SI and CV below the table.

Discussion

Line 282: Am not sure it can be said that SV, SW etc contribute to the XCOM variables? They correlate with them which is different. This is maybe a bigger point that needs discussion and is a concern. The two XCOM variables are correlated with a host of other variables all (or at least many) of which are possibly age related so a correlation is not surprising and not sure what it tells us? If XCOM-G and XCOM P were correlated with height for example there may well be a correlation but that just reflects that both change with age/growth? 

Response: Thank you for your very important remarks. Our previous study (Mani et al., reference [7]) reported that individual gait variables such as SV and SW were not perfectly correlated with age, indicating different developmental characteristics. We thought that by examining the relationships between these five gait functional domains and the straight gait indices could help us understand the kind of control functions that are required for straight gait. 

In this study, we showed that the ability for straight gait develops with age (Table 3) and further showed the gait variables that are associated with two indices of straight gait (Table 4).

Reviewer #2: 

About positioning and rationale of the study

Response: Thank you for your very useful remarks. Certainly, the link between DCD and meandering gait is confusing. So, we decided to remove the content of the first paragraph of the original manuscript from the revised manuscript because it was less relevant to this study. Instead, we clarified the motivation for this study by citing previous studies on the development of gait stability.

The purposes and hypotheses are confusing, particularly purpose 2.

Response: Thanks for pointing it out. It's certainly confusing, so I changed the hypothesis from 3 to 2 (Line 103). Then, I revised the two hypotheses so that they were clearly stated in the discussion (Line 262 and 282).

Methods

The use of the XCOM measures for a whole trial (multiple gait cycles) as opposed to within step appears to be different to the intended use by Hof (2005) and Hallemans et al. (2018), this is not clear within the introduction and has not been fully justified within the manuscript.

Response: Thank you for pointing this out. This study was carried out using a modified method, referring to references 6 and 7. Since there may be an effect of arranging it a little, I added it as a research limit (line 330).

The descriptions of the variables XCOMg and XCOMp describe an integral, however this is not apparent in Figure 1. It looks like you have tried to take a single point in time as an example, but the figure could be improved and/or described properly. Furthermore the ‘Goal axis’ should be added to this figure along x = 0.

Response: Thank you for your advice. It is true that Fig1 is difficult to understand, so I corrected Fig1 and added the explanation of the figure, referring to what you pointed out.

The latter descriptions of these indies on line 277-279 are far clearer than any earlier description so I would suggest using this terminology for the method and the figure to support the understanding of the terms.

Response: Thank you for your advice. Based on your advice, I decided to add the corresponding part to the introduction. (Line 80)

The variables were defined over a period of 2 m with at least 2 gait cycles recorded:

How were these selected from the 6 m walkway?

Due to the evident range in COM position and XCOM over the gait cycle (from figure 1), should the number of gait cycles analysed for each individual have been standardised?

Response: Thank you for your advice. The maximum range for recording all marker data of all participants was 2m. We were concerned that the measurement range (distance) of each subject would change if we used the walking cycle; hence, we decided to analyze by setting the distance to 2 m. This has been described in Line 148. Normalization was done for each individual's COM height.

Results, discussion and conclusions

The Results section is clear with well-presented tables, but would benefit from a statement of results prior to the table introduction.

Again the discussion section would benefit from an introductory sentence or two which summarise the study findings prior to a sub-heading.

Response: Thank you for your useful advice. I added a short heading on line 220 as per your advice.

Line 257: is this supported by both of these variables or XCOMg only?

I don’t think the second and third hypothesis are addressed directly in the discussion.

The statement which concludes your paper: “The two indices presented in this study are effective parameters for evaluating meandering and postural stability during gait”. This feels like it requires further justification and presentation of data. To link with the introduction, gait which is defined as meandering should have been assessed and defined using your two parameters to compare to gait which isn’t meandering.

Response: Thank you for your comment. The first sentence of the discussion indicated two variables. However, I thought the original text was difficult to understand, so as you pointed out, I decided to consider each hypothesis separately. As shown earlier, the hypotheses were changed into two (Line 103), and the discussion was modified to link with the introduction (Line 262 and 283).

Minor comments:

Line 115: suggest ‘reduces to’ 

Response: Thank you for your advice. I changed it as you pointed out (line 104).

Line 117: ‘involved’ doesn’t make sense her as a term

Response: Thank you for pointing this out. As you pointed out, I corrected the sentence to be appropriate for deletion. (Line 105).

119: what are control mechanisms?　

Response: Thanks for your questions. This has been changed to “Control function.” This indicates an element related to postural control during walking, as shown by Mani et al.

Line 132: who provide written consent? The guardian or the child? 

Response: Thank you for pointing this out. The guardians provided consent. This has been added accordingly (Line 120).

Line 158: two gait cycles per trial or in total? 

Response: Thanks for your questions. It is two gait cycles per trial were recorded.

Line 220: I don’t think a % is an appropriate level and this should be reported as a decimal. 

Response: Thank you for pointing this out. We have corrected this accordingly (Line 216).

Line 258: please clearly define which age groups this statement is true for. 

Response: Thank you for pointing this out. We have clearly defined the relevant age group. (Line 263).

Table 1: sex should read ‘male’ or ‘female’ 

Response: Thank you for pointing this out. We have corrected this accordingly (Table 1).

Table 3: check title 

Response: Thank you for pointing this out. I made some changes to the title.

Figure 1 title should define the two factors fully in the title.

Response: Thank you for pointing this out. I have modified Figure 1 and its description.

---

## [Decision Letter · Decision Letter 1]

23 Nov 2022

PONE-D-22-20977R1Developmental changes in straight gait in childhoodPLOS ONE

Dear Dr. Mani,

Thank you for submitting your manuscript to PLOS ONE. After careful consideration, we feel that it has merit but does not fully meet PLOS ONE’s publication criteria as it currently stands. Therefore, we invite you to submit a revised version of the manuscript that addresses the points raised during the review process.

We look forward to receiving your revised manuscript.

Kind regards,

Laura-Anne Marie Furlong

Academic Editor

PLOS ONE

Journal Requirements:

Reviewers' comments:

Reviewer's Responses to Questions

**Comments to the Author**

1. If the authors have adequately addressed your comments raised in a previous round of review and you feel that this manuscript is now acceptable for publication, you may indicate that here to bypass the “Comments to the Author” section, enter your conflict of interest statement in the “Confidential to Editor” section, and submit your "Accept" recommendation.

Reviewer #1: All comments have been addressed

Reviewer #2: All comments have been addressed

2. Is the manuscript technically sound, and do the data support the conclusions?

Reviewer #1: Yes

Reviewer #2: Yes

3. Has the statistical analysis been performed appropriately and rigorously? 

Reviewer #1: I Don't Know

Reviewer #2: Yes

4. Have the authors made all data underlying the findings in their manuscript fully available?

Reviewer #1: Yes

Reviewer #2: No

5. Is the manuscript presented in an intelligible fashion and written in standard English?

Reviewer #1: Yes

Reviewer #2: Yes

6. Review Comments to the Author

Reviewer #1: This sentence in Introduction is not clear-

103 changes in straight gait and the control mechanisms involved. We hypothesized that:

104 (a) straight gait reduces to at younger ages and reaches the adult level around the age

105 of 7 years [4];

is it "ability to walk in straight line is reduced at younger ages"?

Methods

139 the same position. After practicing the task several times, the results were recorded

140 five times

Was a mean of teh 5 trials then used for analysis or one representative?

Reviewer #2: We acknowledge the substantial amendments to the manuscript made by the authors and think that the paper is far improved, but would recommend some further changes and clarification prior to publication.

Introduction: Line 58-59: This still feels a little undefined and not fully justified as a reference to a theme/topic. Also the justification that there is not enough research feels very subjective. To link and lead onto the following paragraph there at least needs to be a description of the relevance of DCD to meandering gait.

Line 86-88: this sentence needs rewording: I am not sure what it is trying to say, it doesn’t make sense.

Line 104: hypothesis 1 needs rewording as ‘reduces to at younger’ does not make sense

Line 128: ‘calling’ would be better described as encouraging them forwards

Line 129: was the walking path painted/drawn on the floor?

Line 160/161: I think earlier this was defined as the position of the researcher? It may be better defined as the start and end point of the walking bouts? This might be better used throughout as opposed to the current reference as ‘start position to parents’ such as line 173 also.

Line 219: these statements feel highly cemented in the outcomes of the statistical tests and without any real consideration of magnitude of effect or what is a meaningful difference. IN Table 3 we can see substantial differences in XCOMG between Adults and 7-8 and 9-10 and I think this should be considered.

Discussion: The XCOMG variable has been discussed with relevance to other data and studies but this has not been compared or contrasted in terms of numerical outcomes in the discussion. In particular Hallemans

Please check the references: I am not sure that they align; Hallemans for example is 6 in the list and 7 in the text.

7. PLOS authors have the option to publish the peer review history of their article (what does this mean?). If published, this will include your full peer review and any attached files.

Reviewer #1: No

Reviewer #2: No

---

## [Author Response · Author response to Decision Letter 1]

22 Dec 2022

Response to Reviewers

We thank you for your peer review and comments on our second submitted manuscript.

We have done our best to address all points raised. Provided a point-by-point response to the reviewer's comments.

In the manuscript, the changed parts are shown in red.

Response to Reviewers:

Reviewer #1: 

This sentence in Introduction is not clear-

changes in straight gait and the control mechanisms involved. We hypothesized that: (a) straight gait reduces to at younger ages and reaches the adult level around the age of 7 years [4];

is it "ability to walk in straight line is reduced at younger ages"?

→Thank you for your advice. We decided the first half of this sentence was confusing, so we fixed it. (Line 114)

Methods

the same position. After practicing the task several times, the results were recorded five times

Was a mean of the 5 trials then used for analysis or one representative?

→Thank you, you are right. We have added an explanation to Line 152.

Reviewer #2: 

Introduction: Line 58-59: This still feels a little undefined and not fully justified as a reference to a theme/topic. Also the justification that there is not enough research feels very subjective. To link and lead onto the following paragraph there at least needs to be a description of the relevance of DCD to meandering gait.

→Thank you for your advice. We discussed the topic of walking in children with developmental disorders, and deepened the content so that the purpose of this research would be linked. (Line 58-69)

Line 86-88: this sentence needs rewording: I am not sure what it is trying to say, it doesn’t make sense.

→Thank you for pointing this out. We fixed the wording. (Line 96-98)

Line 104: hypothesis 1 needs rewording as ‘reduces to at younger’ does not make sense

→Thank you for your advice. We decided the first half of this sentence was confusing, so we fixed it. (Line 114)

Line 128: ‘calling’ would be better described as encouraging them forwards

→Thank you for pointing out. I have corrected it as you pointed out. (Line 138)

Line 129: was the walking path painted/drawn on the floor?

→Yes, you are correct. I added a photo (Fig 1). (Line 157-159)

Line 160/161: I think earlier this was defined as the position of the researcher? It may be better defined as the start and end point of the walking bouts? This might be better used throughout as opposed to the current reference as ‘start position to parents’ such as line 173 also.

→Thank you for pointing out. Sorry, this is the part I forgot to correct in the first review. I decided to define the start and end points of the walk as you suggested. (Line 150-151, 177 and 190)

Line 219: these statements feel highly cemented in the outcomes of the statistical tests and without any real consideration of magnitude of effect or what is a meaningful difference. IN Table 3 we can see substantial differences in XCOMG between Adults and 7-8 and 9-10 and I think this should be considered.

→Thank you for pointing out. Certainly, the values in the 7-8 years group and the 9-10 years group were still small compared to the adult group, which may be due to the growth process. However, it is true that there were no statistically significant differences between the 7-8years group compared to the adult group. So I mentioned it in the discussion, not the result.

Discussion: The XCOMG variable has been discussed with relevance to other data and studies but this has not been compared or contrasted in terms of numerical outcomes in the discussion. 

→Thank you for your advice.Based on the results of comparison by age, we added what kind of abilities can be acquired by what age (Line 306-309).

In particular Hallemans

Please check the references: I am not sure that they align; Hallemans for example is 6 in the list and 7 in the text.

→Thank you for pointing this out. We rechecked and revised the reference list and manuscript.

---

## [Decision Letter · Decision Letter 2]

17 Jan 2023

Developmental changes in straight gait in childhood

PONE-D-22-20977R2

Dear Dr. Mani,

We’re pleased to inform you that your manuscript has been judged scientifically suitable for publication and will be formally accepted for publication once it meets all outstanding technical requirements.

Kind regards,

Dimitrios Sokratis Komaris, Ph.D

Academic Editor

PLOS ONE

Additional Editor Comments (optional):

**Both reviewers have made editorial comments, kindly review them below and make appropriate corrections before submitting your final manuscript.**

**Discussion; page 16; line 305: "Thus, this index expresses one of the ability to walk straight." This is not clear. Would/should it be better written as: "Thus, this index expresses one COMPONENT of the ability to wall straight."**

**And**

**Line 66 remove or reword ‘it shows’**

**Figure 3 add years to x axis.**

Reviewers' comments:

Reviewer's Responses to Questions

**Comments to the Author**

1. If the authors have adequately addressed your comments raised in a previous round of review and you feel that this manuscript is now acceptable for publication, you may indicate that here to bypass the “Comments to the Author” section, enter your conflict of interest statement in the “Confidential to Editor” section, and submit your "Accept" recommendation.

Reviewer #1: All comments have been addressed

Reviewer #2: All comments have been addressed

2. Is the manuscript technically sound, and do the data support the conclusions?

Reviewer #1: Yes

Reviewer #2: Yes

3. Has the statistical analysis been performed appropriately and rigorously? 

Reviewer #1: Yes

Reviewer #2: Yes

4. Have the authors made all data underlying the findings in their manuscript fully available?

Reviewer #1: Yes

Reviewer #2: Yes

5. Is the manuscript presented in an intelligible fashion and written in standard English?

Reviewer #1: Yes

Reviewer #2: Yes

6. Review Comments to the Author

Reviewer #1: Thank you for the work in addressing comments to date and eh manuscript is significantly improved and clear.

**I have one remaining query regarding a sentence which is unclear and needs to be clarified:**

**Discussion; page 16; line 305:**

**"Thus, this index expresses one of the ability to walk straight."**

**This is not clear. Would/should it be better written-**

**"Thus, this index expresses one COMPONENT of the ability to wall straight."**

Reviewer #2: The authors have responded to the revisions and added further description to the introduction which helps identify the rationale for the work and other research in this field. The method is now also clearer with further description and a figure which supports the interpretation of the text and helps reproducibility.

**Minor comments which could be amended to support the readability:**

**Line 66 remove or reword ‘it shows’**

**Figure 3 add years to x axis**

7. PLOS authors have the option to publish the peer review history of their article (what does this mean?). If published, this will include your full peer review and any attached files.

Reviewer #1: No

Reviewer #2: No

---

## [Editor Report · Acceptance letter]

31 Jan 2023

PONE-D-22-20977R2 

Developmental changes in straight gait in childhood 

Dear Dr. Mani:

I'm pleased to inform you that your manuscript has been deemed suitable for publication in PLOS ONE. Congratulations! Your manuscript is now with our production department. 

Kind regards, 

on behalf of

Dr. Dimitrios Sokratis Komaris 

Academic Editor

PLOS ONE